# A Conceptual Framework for Empowerment of Psychiatric Nurses Caring for Children with Mental Health Challenges

**DOI:** 10.3390/ijerph22030396

**Published:** 2025-03-07

**Authors:** Rorisang Mary Machailo, Magdalena Petronella Koen, Molekodi Jacob Matsipane

**Affiliations:** School of Nursing, Faculty of Health Sciences, North West University South Africa, Potchefstroom 2531, South Africa; dr.d.koen@gmail.com (M.P.K.); molekodi.matsipane@nwu.ac.za (M.J.M.)

**Keywords:** conceptual framework, children, empowerment, mental health, psychiatric nurses

## Abstract

There are scanty empirical and theoretical studies focusing specifically on the implementation models for the care of psychiatric nurses caring for children. Staff contribution in the process of putting clinically relevant actions into practice contributes to effective implementation, increased acceptance, and commitment. These practises can be used to improve mental health care services of children at different levels. The aim is to deepen an understanding of the perspective of care for children with mental health challenges. A qualitative, exploratory, and descriptive design was used to allow for innovative thoughts to restructure the practice of child psychiatric nursing and is based on the Practice-Orientated Theory of Dickoff. A positive focus on aspects that facilitate care for children with mental health problems is informed by the data collected. A suite of services that include practice environment, trainings, and practical assistance is feasible to support psychiatric nurses. This framework implies that psychiatric nurses need professional competence to understand the context of the environment. Psychiatric nurses need to move beyond engagement to demonstrate how they help children with psychiatric challenges and to enable the development of psychiatric nursing practice through international collaboration.

## 1. Introduction

Globally, the demand for child mental health services continues to grow and this challenge has become a growing concern. Clinically effective practises can be used to improve mental health care services of children at different levels like on the individual patient, appropriate programmes, organisations, and system levels [1]. At the same time, staff participation in establishing relevant procedures into practice contributes to the effective implementation and increased uptake of these developed practises. Shortages in child psychiatric nursing workforce have been observed for years, and the urgency of integrating the psychiatric nurse practitioner within the clinics and health care centres must become a priority that capitalise on their clinical expertise [2].

Whilst SDG 3 (Sustainable Development Goal 3) focuses on ensuring healthy lives and promoting wellbeing for all at all ages, the literature highlights that the quality of psychiatric services for children is substandard across both health and social systems. Health systems have deficits in care competence, system competence, user experience, service provision for common and serious conditions, and service offerings for adolescents [3]. Therefore, there is an urgent need for strong nursing leadership and innovation in health care in a time of rising rates of childhood mental health challenges. This is not only a professional obligation, but a moral one. Psychiatric and mental health nurse practitioners have the opportunity, as well as the ethical and professional obligation, to play a leading role in improving child and adolescent mental health [4].

Psychiatric nursing is defined as a speciality within the nursing profession that is concerned with prevention, care, and the rehabilitation of mentally ill patients. Psychiatric nurses assess patients through interviews, observation, and interaction. They assist in restoring the emotional wellbeing of patients through the provision of individual counselling to both patients and families to assist those in their care in understanding the illness [5]. Psychiatric nurses possess unique clinical and interpersonal skills for working with children with mental illness and this practice environment is characterised by mammoth challenges in practising effectively due to the dynamics thereof [6].

It is known in the health sector that critical human developmental changes take place within the first 18 years of life. Psychiatric nurses fill the gap to care for this vulnerable population [7]. A child’s cognitive development, social, and emotional characteristics are all influenced by their childhood contexts [8]. Children are in a particularly critical period for the identification, early intervention, and treatment of mental health challenges [9]. Psychiatric challenges during childhood may be attributed to negative experiences, and potentially stressful events experienced before the age of 18 years. These may include but are not limited to forms of abuse, behavioural problems, psychological and social wellbeing, neglect, and emotional disturbances which entails household dysfunction. Previous research confirms that adverse childhood exposure is linked to poorer health-related quality of life [10].

Despite the extent and importance of mental health problems in children, most researchers have consistently established a treatment gap where there is a lack of a framework and strategies to enhance psychiatric nursing care of children [9]. In addition, empirical research that focuses specifically on the implementation of models for care for psychiatric nurses is scarce. The available literature often lists factors that influence implementation without proposing how effective implementation takes place. It also highlights that despite growing evidence of effective interventions, health and social systems fail to implement the interventions to address the mental health needs of children. These deficits in the quality of health systems are not favourable to children [3].

Psychiatric nurses ought to take family into account and seek to involve them in treatment planning. The literature illustrates that child- and family-centred care practises increase patient and family satisfaction, decrease child and parent anxiety, facilitate more rapid recovery, have a positive impact on the mental health of caregivers who have children with chronic illness, and increase staff satisfaction [11]. Psychiatric nurses have historically been innovators in mental health care and delivery systems of care. It is also important to recognise that child and adolescent psychiatric mental health nursing has its roots in social justice [7].

The South African Nursing Council (SANC) [12] defines a Mental Health Nurse as a health care practitioner who is a professional nurse trained as a mental health care nurse specialist and can provide prescribed mental health care, treatment, and rehabilitation services. Such a nurse holds an additional qualification in Mental Health Nursing. Even though nurses may be trained to the scope of their practice, it is critical to understand that external barriers such as opposition from other health professions impede full practice from occurring [7]. Therefore, the authors of this article postulate that a widespread promotive, preventive, and recovery-oriented framework could be useful to psychiatric nurses caring for children. These would make them become experts imbued with the contextual knowledge and clinical skills to work competently with mentally ill children [13].

## 2. Materials and Methods

### 2.1. Study Design and Process

A qualitative, exploratory, descriptive, and contextual design was used to allow for innovative ideas that can restructure the practice of child psychiatric nursing [14]. The framework was developed from the themes and central concepts that were identified from data collected through five focus group discussions and a review of the literature.

### 2.2. Sampling and Data Collection

A literature review was conducted because it is recognised as an important source of rigorously and transparently synthesised information. The initial search yielded 273 articles, of which 53 results appeared to be candidates for inclusion in the review and were subject to full-text screening. In the end, only four studies qualified for inclusion and synthesis. Five focus group discussions which lasted approximately an hour were conducted. The participants were psychiatric nurses in a child psychiatric practice environment.

### 2.3. Data Analysis

The theory developed by [15] Dickoff, James, and Wiedenbach (1968) was used to generate a framework to empower psychiatric nurses caring for children with mental health challenges in NWP [16]. The framework combined data collected through focus groups and a review of the literature. These concepts were contextualised to formulate this conceptual framework. This conceptual framework explains concepts and analyses the prescribed actions that inform attaining the activities.

### 2.4. Ethical Approval

Ethical approval to conduct this study was obtained from the Scientific Committee of the School of Nursing Science and the North-West University Health Research Ethics Committee (HREC Reference Number: NWU-00278-21-A1). This study is part of the doctoral research of the first author and the empirical phase of the degree. Permission to conduct this study was requested and obtained from the North-West Provincial Department of Health, directors of the district health services of NWP, and the Chief Executive Officers of accredited mental health care institutions. Permission was obtained from participants, and they signed informed consent forms before taking part in this study.

## 3. Results

The conceptual framework explains concepts and analyses the relevant activities that inform attaining the strategies. This framework follows objectivist deductive research. This approach builds knowledge by developing increasingly better understandings of, and insights into the workings of the real world [17]. Figure 1 presents the elements adapted from the theory of [15] to illustrate a conceptual framework and process for developing a guide for psychiatric nurses in caring for children. Each constituent forms part of the discoveries from the preceding studies. These studies included the literature review and appreciative inquiry which assisted with the development of this framework. The characteristic for elements such as agent, recipients, context, guiding process, energy source, and the product are discussed, contextualised, and described below.

### 3.1. The Agent

The agent refers to the psychiatric nurse responsible for planning, coordinating, and caring for children in child psychiatric practice. This nurse has characteristics and traits like self-awareness, empathy, and is capable of initiating care for children in need of psychiatric services [16]. According to [17], the agent ought to bring about positive changes, evaluate the clinical practice to identify areas in need of enhancement that would strengthen the efficiency, improve job satisfaction, and improve patient outcomes. Psychiatric nurses have unique competencies that they bring to the practice. They collaborate with the mentally ill patients to comply with the requirements of treatment to live a rewarding life [5]. These nurses are warm, caring, and non-judgmental [16]. Psychiatric nurses provide key psychiatric care in both governmental and non-governmental subdivisions. These different work locations and broader scope of practice, including the provision of both psychotherapy and treatment management, give these consultants a unique capacity to address boundaries in accessing health care services. The psychiatric nurse practitioner’s role emphasises on assessment, diagnosing, and the psychopharmacology of individuals across the lifespan [7].

#### 3.1.1. Empowering Abilities

Both the literature review and focus group interviews highlighted the need for empowerment. The psychiatric nurse as the agent should perform empowering activities to enhance knowledge, skills, and abilities to enable recipients to address their strengths and weaknesses [18] The following quote supports these abilities:


*P1FG3: ‘So maybe just the education to the public about, what we are, who we are and what we actually do. And also, the parents to understand the condition of the child. Like when we say the child is better, what is it that the child can do’.*


The literature highlights that, once empowered, psychiatric nurses release the energy within them using knowledge, experience, and motivation to create a healthy work environment specifically directed to achieve positive outcomes for the children [19].

#### 3.1.2. Self-Awareness

Self-awareness is the mental act of investigating ourselves while being motivated by a healthy inquisitiveness into who we are. Self-awareness does not represent a unitary proposition, but self-reflection which is associated with positive outcomes, self-regulation, and self-knowledge. These involve becoming the person one wants to be and include knowing who you honestly are [20].

Therefore, the greater the psychiatric nurse’s self-awareness, the greater the ability for their resemblance, acceptance, and empathy [17]. Empathy is an expertise in psychiatric nursing practice involving self-awareness and the use of emotion in relational understanding.

#### 3.1.3. Initiative

The psychiatric nurse ought to take the initiative to go beyond problem spotting to problem solving and thereafter articulate opportunities that exist for improvement. These include the observation and clarification of roles and expectations. A suite of services including guidelines covering practice environment, trainings sessions, and technical assistance is feasible to support psychiatric nurses [21].

### 3.2. The Recipients of the Conceptual Framework

The recipient is described as those receiving the activity [16]. These are children with mental illness receiving direct improvement from participating in the implementation process obtaining knowledge, skills, and abilities to cope. The following quote confirms the awareness of psychiatric nurses with children:


*FG3P4: ‘Okay with children sometimes I understand it because it is difficult to diagnose them when they are young. Because children are active naturally. You are going to say they are hyperactive when they are not because they start settling when they go to school. And sometimes it is difficult to say the problem might be the environment. So, you correct the environment then the child will be fine’.*


The literature confirms that the onset of mental health encounters in children often occurs at a time in life that is distinguished by considerable instability. These may include moving home, changing school, being bullied, witnessing domestic violence, being abused, parents separating or divorcing, initial experiences with alcohol and drugs, as well as many other factors. These factors contribute to children having difficulty with coping with their age-appropriate development and may have serious, long-term consequences for quality of life. Their living circumstances are often subject to complex psychological assessment and processing viewed as multidimensional constructs comprising a physical, emotional, and social component of wellbeing and function [22]. Whilst the ability to regulate emotions develops rapidly in the early years of life, this improves slowly into adulthood. Emotional regulation is a process of recognising emotions and using strategies to alter emotions to achieve a desired outcome [23]. The psychiatric nurses ought to understand the child’s emotional development so that they can assist.

#### 3.2.1. Cognitive Development

The focus group discussions highlighted the need for understanding the normal development of a child. Nurturing and responsive relationships during the early years of a child’s life are integral in the development of cognitive and language skills. Researchers highlights that the childhood-onset of mental illness affects individuals while cognitive development is still ongoing, and it is more likely to have lasting deleterious effects based on residual memory impairment [24]. At the same time, [25] postulates that children need scaffolding to engage and progress in active and effective collaborative learning interactions. Therefore, it is crucial that psychiatric nurses understand the cognitive development of a child so that they can know when to act.

#### 3.2.2. Safety

Physical and emotional safety are important to the care of a child. Both the treatment and home environments must be psychologically and physically safe for children. Safety requires checks, policies on visitors, and tracking and monitoring systems to avert risks, incidents, and safety breaches [26]. The focus group discussions raised the following point:


*FG1P3: ‘Children are naughty, they will be moving around. They can also have injuries things like that. We also need to be observant and like supervise them all the time. So, enough staff is needed.’*


According to [27], safety advocates that there should be synergy between both professionals and institutions to establish a safe space for mental health care users. This space must be sensitive and responsive to their social, political, linguistic, economic, and spiritual concerns [28]. Therefore, psychiatric nurses ought to reflect on their own practises regarding the safety of the children in their care.

### 3.3. The Context of Psychiatric Nursing Care

The theory of [15] Dickoff, James, and Wiedenbach (1968) defines this aspect as the context where the activity takes place [16]. In this study, the context is the health care facility providing child psychiatric and mental health care services. Children and adolescents (C&A) constitute one-third of the world’s inhabitants and half of the population of low- and middle-income countries. However, mental health resources for these populations are rare, insufficient, unevenly distributed, and inadequately used, with effective and culturally relevant treatment rarely available. Preventive and curative care is often substandard, with low diagnostic meticulousness, treatment delays, and non-adherence to evidence-based values and quality is often worse for low-income communities [3]. The discussions in the group elicited the following responses:


*FG1P8: ‘Most of the treatment of children with mental health issues you won’t find it at the clinic. They need a special prescription. So, finding them at the clinic is very difficult.’*


Child mental health care infrastructure is limited by factors such as geography, culture, lack of cost-effective research, and poor intersectoral collaboration [28]. Investment into clinical and scientific infrastructure is an important factor to increase access to sustainable care [29]. The important infrastructural components include playrooms and relevant toys, play areas, appropriate furniture such as tables and chairs, beds, toilets, computers, internet access, and wireless connectivity including the integration of systems and IT support onsite [30]. Children in acute distress need an environment that assures both safety and healing [28]. The staff in this environment has a split focus and these are as follows: attending to the individual child and then the overall environment of care.

#### 3.3.1. Legal and Policy Framework

Children and adolescent mental health problems are related to each country’s development and global changes. National policies must be designed to ensure that children can access even the most basic mental health care facilities [31]. In this study, both the literature review and focus group discussions confirmed that there is no compliance with the legal framework of the practice environment. The following vignette confirms the deficiencies:


*FG2P1: ‘We need improvement. We need systems in place because currently the district mental health specialist teams have not been fully implemented as mandated by the policy guidelines. And that policy guideline expired in 2020, but three years after the expiry of five years, it is still only the psychologists within the team for the whole mental specialist team. And remember we need to have a psyche nurse, the psychologist, the psychiatrist and others.’*


The global demand and emphasis on nurses’ involvement in health policy and systems development has increased over the years. However, difficulties still exist in implementing various policies and related activities due to the lack of cooperation and coordination among different provinces, health care institutions, and private organisations [24]. Policies and regulations that promote mental health must be developed [31]. Psychiatric nursing practice does not have the autonomy to develop its infrastructure. Active participation by policy makers and the Department of Health’s management such as financing are necessary. This would be supported by the actual participation of psychiatric nurses in health policy reforms. Therefore, the advocacy voice of psychiatric nurses can only be heard if they participate in policy formulations and reforms.

#### 3.3.2. Supportive Environment

This study established that caring for children with mental health challenges is complex and can also be overwhelming for the inexperienced psychiatric nurse as a provider. The child psychiatric and mental health setting affirms the advantages of utilising interdisciplinary teams in the delivery of care [32].

It has been clear that psychiatric nurses need support which includes the following: training to alleviate psychological stress, teams support to augment trust, dissemination of knowledge to facilitate clinical decision making, support and recognition from the health care team, government, and community [33]. Psychological interventions aimed at enhancing resilience in the individual may be of benefit if contextualised and environmental factors are addressed.

### 3.4. The Guiding Procedure

This study recognised that psychiatric nursing practice is a complex interpersonal and intrapersonal process that requires the professional nurse to be fully competent. This competence must be incorporated into a systematic process of care, which demands the integration of theory and practice including an in-depth understanding of the complexity of children who are mentally ill. Interpersonal interactions build the trusting relationships which are the foundational blocks for the communication, psychoeducation, and prevention of the consequences of unacknowledged social media, stigma, and mental health problems [7]. Psychiatric nurses need to be interpersonally competent before engaging with children who are mental health care users [34]. Competence is an array of abilities, including the ability to seek and adapt to new and relevant information for clinical decision making that develops through clinical experience [35]. Therefore, psychiatric nurses must be adaptive and find a purpose in understanding the cultural factors which play a role in the given moment.

#### 3.4.1. Capacity Building

Training and skill development enable psychiatric nurses to identify and address emerging problems before they meet diagnostic criteria. Specific guidance on early childhood mental health may be beneficial [29]. This is affirmed by the following quote from the focus group discissions:


*P2FG3: ‘Advanced psyche will be beneficial to us. But nowadays it is scares. You can only find it in far cities and not sure of which universities because the curriculum is changing’.*


Whilst clinical teaching and development enhance clinical skills and prepare the psychiatric nurse to work in a systems-based practice [32], increasing the capacity of these nurses as mental health care providers to address practical, logistical, and emotional barriers to patient’s engagement like children in clinical practice is crucial [29]. The literature advocates for the design and implementation of socially and culturally sensitive education. These include capacity building strategies that include training, support, and supervision for all screening for mental illness in children [24]. Therefore, continuous professional development is central in determining a suitable and enabling environments for all nurses in child psychiatric practice.

#### 3.4.2. Management Support

The literature highlights the need for and interest in designing a programme that would address the unique needs of nurse managers and subsequently impacts the quality of patient care which is influenced by the level of this leadership [36]. Psychiatric nurse managers should be encouraged to participate in leadership and management training programmes. These programmes allow for the enhancement of the global understanding of leadership panaches, develop a relationship with organisational leaders, and attain awareness into how their environment internally operate [37]. The need for an increased emphasis on manager engagement, flexibility, collaboration, crossing boundaries, and collective leadership has been identified [36]. Nurse managers’ role is leadership, and such managers are responsible for the health of their staff. The nursing staff rely on them for emotional support [26]. Nurse managers need to be empowered in supporting the staff and encouragement, such as searching for little things that they could do for them, including emotional support [38]. This becomes an empowerment process of unleashing the power in them and their knowledge, experience, and motivation, including focusing that power to achieve positive outcomes for the children.

### 3.5. Energy Sources

These are the resources invested in accomplishing psychiatric nursing care. The concepts of professionalism, skills, and knowledge are factors that focus on the capability of the psychiatric nurse to satisfy the needs of the children [39]. The elements thereof are described and contextualised below.

#### 3.5.1. Skills and Knowledge

A psychotherapeutic relationship is a defining element of psychiatric nursing practice. In psychiatric nursing, major and essential knowledge include skills related to the intrapersonal and interpersonal dynamics within a person [16]. The following quote confirms these skills and knowledge:


*FG4P6: ‘The interesting part is that at least we have nurses who have done child psyche’.*


Psychiatric nurses are trained in multiple sciences ranging from psychiatry, medical science, psychotherapy, neuroscience, to prevention science. The organising platform for their practice is intervention within the context of relationships. Inpatient psychiatric nurses often form the present moment in response to a child experiencing an acute exacerbation of mental illness [28]. Patterns of building and maintaining relationships with a child within the family crucially contribute to psychiatric nurses’ knowledge, skills, confidence, and engagement.

#### 3.5.2. Empowering Environment

Empowerment is rooted in social action and aims to increase the autonomy, power, and influence of burdened groups. The World Health Organisation (WHO) introduced the new public health strategy in the Ottawa Charter. One of the main principles is the redistribution of power from health professionals to health care service users. In this instance, user participation, involvement, and health competency gain increased attention in health care and research [18]. Increasing the capacity of psychiatric nurses as mental health care providers to address practical, logistical, and psychological barriers to patient engagement in mental health care is important [29]. Psychiatric nurses need to be prepared for this environment as adolescents, in particular, value accessible services with the following empowering abilities: friendly and respectful providers, clear communication, competent health care, confidentiality, age-appropriate environments, involvement in decisions, and positive health outcomes [3].

#### 3.5.3. Research and Information

Research provides insight into the extent of the problem and assists in identifying gaps in the practice environment. It also reflects improvements in service delivery and outcomes. Both the empirical and theoretical phases of this study proved that nursing research is fundamental for the profession and necessary to promote innovative quality care thereby influencing health policies [40]. Focus group discissions highlighted the need for research.


*FG2P2: ‘We don’t know, maybe because of this type of research that you are doing, maybe you can help us to some extent.’*


Investing in nursing research and infrastructure allows staff to conduct research and transfer evidence-based knowledge into practice, especially where the hospitals and clinics have neither the capacity nor the infrastructure for research-based activities. The quality of research infrastructure qualitatively affects the development of nursing practice [41].

### 3.6. The End Product

This step focuses on achieving an outcome that indicates the increased competence and confidence of psychiatric nurses in caring for children in clinical practice. These may include improved communication, creative ways of care, and achieving quality of life through changed social and family relations [39]. The concepts and elements that support the creative and analytical thinking of psychiatric nurses to enhance the quality of care within the child psychiatric environment are discussed below.

#### 3.6.1. Quality of Care

Quality of care is achieved when the appropriate service is provided in an efficient manner and the desired outcome is reached [42]. Both empirical and theoretical phases of this study confirmed that the quality of health care services and adequacy thereof can be measured based on the views and satisfaction of those receiving care. Total quality management in a health care system includes professional knowledge, competence including lived experiences of patients about the type and level of care received. In recent years, there has been a shift towards a patient-centred measure of gratification with the quality of nursing care received. This is a major component of a health care institution’s quality management systems currently [43]. Therefore, the quality of psychiatric nursing care offered to children could be improved through the concepts discussed below.

#### 3.6.2. Caring Standards

Standards in health care facilities are declarations that describe the required key functions, activities, processes, and structures so that various departments in a facility can provide quality services. The office of the health standard compliance was established to ensure that both public and private health institutions in South Africa comply with the required health standards. This office monitors and enforces health care safety and quality standards in health establishments. Focus group discissions clarified the need for standards of care through the following quote:


*FG4P2: ‘You know it is supposed to be Multi-Disciplinary Team (MDT). Each one of the MDT personnel needs to see the patient. From there the MDT team sit and present the case. They discuss and come up with the diagnoses. After the discussion and all the underlying whatsoever. And then the planning of treatment. But it is not done like that here.’*


Competence, which is affirmed by an array of abilities, including the ability to seek and adapt to relevant new information for clinical decision making, develops through clinical experience [35]. Competencies are the specific observable and measurable abilities that are expected within each domain of practice and for each population. It is complex to define [44]. Proficiency, which is the last stage of development in any clinical field and treatment of mental illness takes years of self-reflective learning and practice [35]. Cultural competence is a critical component of child and family centred care. To achieve healing the patient must make life affirming choices from within the context of their own culture [45]. Psychiatric nurses give nursing care to an increasingly diverse patient population and the need to be culturally competent is supreme.

#### 3.6.3. Teamwork and Collaboration

Teamwork encourages knowledge sharing and vision. Vision in teams is goal-driven and task-orientated. When team goals and tasks require mutual interdependence among members, knowledge sharing becomes desirable because it removes barriers to outcomes. Knowledge sharing raises the ability of a team to work toward common end [46]. The participants in focus group discussions confirmed this:


*FG1P5: ‘I think having nurses with different categories and exposure or how do I say it. Having different experience and specialities yes. Because if there are nurses who are good in medical, they can assist with the medical part of it. Those who are good or having a speciality in surgical, they can help with it. And those that are good with mental, they can also help’.*


Collaborative knowledge sharing was affirmed through group discussions and indicated that effective communication among health care professionals is critical to appropriate the continuity of care. At the same time, displaying leadership and trust is essential to make the rest of the health care professionals understand nurses’ skills [40]. The collaborative care model that emphasises collaboration between mental health care providers and patients is patent in the literature, with mental health providers affirmed more efficiently by researchers [29].

## 4. Discussion

The conceptual framework presented in Figure 2 is recommended to address the gap existing in previous studies, which lack guidance to enhance nursing care for children who are mentally ill. As psychiatric nursing is concerned with the application of psychiatric principles and nursing care, this framework supports and build on interpersonal and therapeutic communication, including relationship development. The literature affirms psychiatric nurses as experts in crisis intervention, mental health assessment, medication and therapy, and patient assistance. This framework may be used in different clinical settings like the outpatient department where children are treated, in the wards where patients are admitted, and may need to be modified to suit parents when in the community.

With the growing demand for mental health services and associated high costs to health care systems, there is an urgent need for cost-efficient delivery approaches, particularly for youth and remote populations. There is a need to emphasise spending wisely for better value for money. On the other hand, taking care of severe and acute child mental health conditions is imperative. Therefore, investing in child mental health is important for the generation to come. This also supports the Sustainable Development Goal 03 of the United Nations. Therefore, policy makers are to ensure effective systems of care that require investments matching with needs of society, and allocate resources based on cost-effectiveness, impact, and equity financing [47]. The literature affirms that most mental health problems and illnesses begin in childhood or adolescence, and early interventions delivered through primary prevention and primary care can prevent delays in receiving care, and thereby also yield significant returns on investment [48].

Part of the caring standards that will enhance child mental health care is to develop recovery programmes and community-based services that empower the service user to take the lead in their own treatment. Community mental health centred on rural areas are often underserved due to the shortage of mental health providers and limited resources to children. The initiative of a psychiatric nurse will be useful in such situations. Whilst a rural psychiatric nurse may encounter several barriers to providing adequate mental health care to children, this psychiatric nurse must understand rural culture, values, and beliefs in these communities [49]. This is also in line with African epistemologies on child development that guide parents in child-rearing patterns. In this instance, there is collective discipline and support for difficult children. Home environments and cultural backgrounds play an important role in the mental health of every child [50].

Effective communication and collaboration among health care professionals are critical to the appropriate continuity of care [40]. To achieve the goals of the mental health system, a work force is required. Therefore, psychiatric nurses ought to have expert knowledge, be competent with a degree of empowerment, coupled with high levels of attachment and commitment, including willingness to become involved in the activities beyond their common and pre-determined duties [51].

The therapeutic relationship is crucial in the practice of mental health nursing. The purpose of the therapeutic relationship is to give support through interpersonal communication that makes it possible to understand the needs and perceptions of the person and empower them for self-management. The attitudes of psychiatric nurses that support the therapeutic relationship comprise the following: showing understanding and empathy, accepting individuality, giving support, being accessible, being genuine, promoting equality, showing respect, maintaining clear boundaries, and having self-awareness [52]. Psychiatric nurses need to relate meaningfully to children’s reactions to their illness, including the psychological and social changes that illness forces on them. Children may react to their mental distress by normalising the experience or projecting their problems onto the perceived deficiencies of caregivers. Therefore, psychiatric nurses need to articulate the psychotherapeutic processes operating within the engagement and relationship building processes so they might focus on shared emotional state [28].

The role and scope of nursing practice have evolved in response to the changing needs of individuals, communities, and health services [53]. At the same time, health systems are crucial for children’s health and wellbeing but need to be augmented by social systems to reach this age group with promotive, preventative, and curative services relevant to their life stage [3]. Primary health care is an important source of essential health care, including for underserved populations. Such care could be a powerful platform for responding to a range of health challenges in developing countries, but only if we give increased attention to the infrastructure and supplies at the health centres and clinics that serve as the first line of care for children with mental health challenges, particularly in rural areas [54].

## 5. Conclusions

This conceptual framework contributes to the enhancement of the mental health care services of children and empowers psychiatric nurses in improving the quality of life of children who require support. Allocating funding to support the scaling-up of cost-effective mental health services to children will strengthen the management and evaluation of the strategies being implemented. This will assist in the evaluation of the outcomes and priorities set for children. Future longitudinal studies could explore whether the exposure to this framework contributes to changes over time as psychiatric nurses gain more experience. There is a need for research on local social and cultural issues that affect the mental health of children, the integration of mental health care services into primary health care, and the scaling-up of programmes which would assist in the implementation of this framework.

## Figures and Tables

**Figure 1 ijerph-22-00396-f001:**
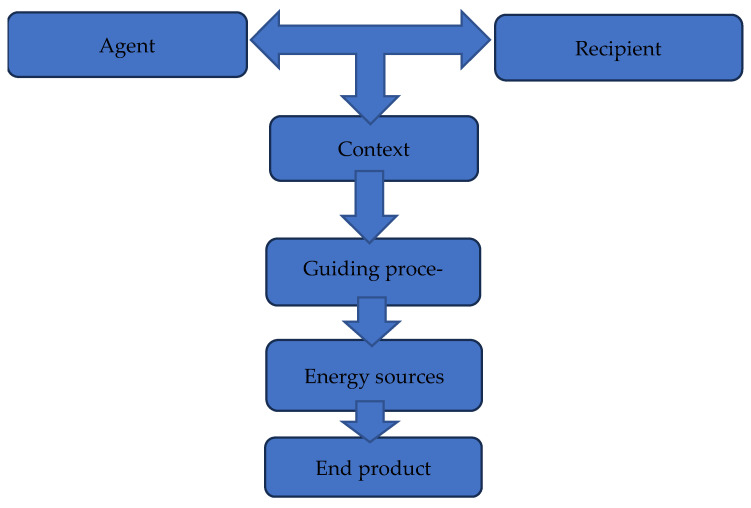
Reasoning map.

**Figure 2 ijerph-22-00396-f002:**
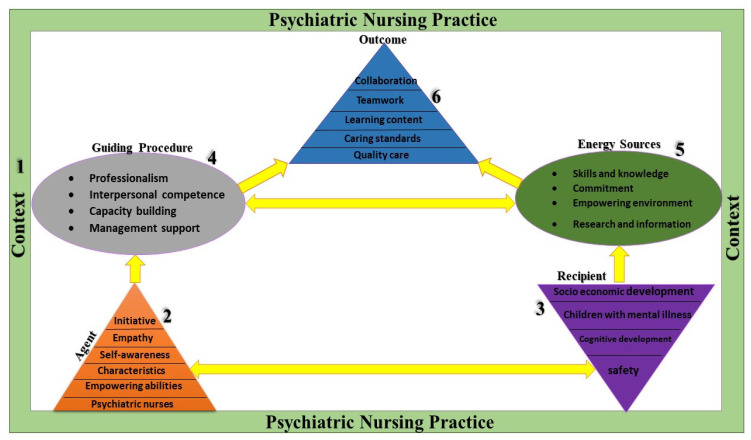
The empowerment framework for psychiatric nurses in child psychiatry.

## Data Availability

The data presented in this study are available on request from the corresponding author due to ethical reasons.

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
