# Peer review of "A Conceptual Framework for Empowerment of Psychiatric Nurses Caring for Children with Mental Health Challenges"

_ijerph, 2025, doi:10.3390/ijerph22030396_

Round 1
Reviewer 1 Report
Comments and Suggestions for Authors
This is an interesting article however, it needs some improvements.
In the abstract, avoid using phrases like "clinically relevant measures" repeatedly. Instead, use clear terms that show what is new about the study.
The introduction should focus on the importance of fixing issues in mental health care for children. However, there is too much repetition about the need for better child mental health nursing care. Make these parts shorter to avoid sounding too strong. Use newer global or regional data to show how urgent the problem is.
The "Materials and Methods" section is not clear and needs to cover all parts, such as objectives, method selection, sampling, research process, data analysis and ethics.
In the Results section, make the information easier to understand so it's more readable.
For example, Figure 1 (Reasoning Map) is mentioned but not well explained in the text. Describe its parts more clearly, especially how they work together to create the framework.
The Results often repeat familiar ideas (like empowerment and self-awareness). Focus on new insights that come from the data.
For each part of the framework, give specific examples or quotes from the people involved.
The discussion just repeats the findings instead of giving a deeper analysis. Talk about how the framework fills in missing parts of what we know.
Think about the framework's weaknesses and how well it can be used in different healthcare places.
Even though the framework is suggested, it's not clear how it can be used in real life. For example, how can hospitals or training programs actually use these ideas?
The conclusions should detail the specific, measurable outcomes this framework is designed to achieve, such as nurse retention or improved patient outcomes. Provide more active recommendations for researchers and policymakers
See again referencing style for consistency; at times, "et al." is not included.
You can also improve language. For example, avoid passive constructions, define terms like "psychological stress", "sufficiently empowered" and make sure that any table, figure, etc. in your paper is well-captioned and described in the text.
Comments on the Quality of English LanguagePlease make a thorough text editing and work with the academic writing of the article.
Author Response
|
Response to Reviewer 1 Comments
|
||
|
1. Summary |
|
|
|
Thank you very much for taking the time to review this manuscript. Please find the detailed responses below and the corresponding revisions/corrections highlighted/in track changes in the re-submitted files.
|
||
|
2. Questions for General Evaluation |
Reviewer’s Evaluation |
Response and Revisions |
|
Does the introduction provide sufficient background and include all relevant references?
|
Yes |
|
|
Is the research design appropriate?
|
Can be improved |
|
|
Are the methods adequately described?
|
Can be improved |
|
|
Are the results clearly presented? |
Can be improved
|
|
|
Are the conclusions supported by the results?
|
Yes |
|
|
3. Point-by-point response to Comments and Suggestions for Authors |
||
|
Comments 1: In the abstract, avoid using phrases like "clinically relevant measures" repeatedly. Instead, use clear terms that show what is new about the study.
|
||
|
Response 1: Thank you for pointing this out. We agree with this comment. Therefore, we have rephrased’ clinically relevant measures on sentence 11 with ‘practices’
|
||
|
Comments 2: The introduction should focus on the importance of fixing issues in mental health care for children. However, there is too much repetition about the need for better mental health nursing care. Make these parts shorter to avoid sounding too strong. Use newer global or regional data to show how urgent the problem is.
|
||
|
Response 2: We have accordingly revised the introduction with a focus on preventative and promotive context on line 91-93.
Comments 3: The "Materials and Methods" section is not clear and needs to cover all parts, such as objectives, method selection, sampling, research process, data analysis and ethics.
Response 3: Thank you for pointing this out. We have revised the material and methods section from line 97 to line 126. We have added and discussed sub-heading study design, sampling, data collection, analysis and ethics
Comments 4: In the Results section, make the information easier to understand so it's more readable. For example, Figure 1 (Reasoning Map) is mentioned but not well explained in the text. Describe its parts more clearly, especially how they work together to create the framework.
Response 4: In objectivist deductive research, a conceptual framework typically includes a description of relevant literature, a summary of the relevant theory, an explanation of why this theory could be informative to this context. This has been described on line 130 to 132. Each part has been described from line 161 to 428 and each heading is italic.
Comments 5: The Results often repeat familiar ideas (like empowerment and self-awareness). Focus on new insights that come from the data. Response 5: The results are based on the research themes of focus groups and literature review
Comments 6: For each part of the framework, give specific examples or quotes from the people involved
Response 6: We have accordingly revised Quotations and have made additions
Comments 7: The discussion just repeats the findings instead of giving a deeper analysis. Talk about how the framework fills in missing parts of what we know. Think about the framework's weaknesses and how well it can be used in different healthcare places
Response 7: Thank you for pointing this out. The discussion has been reviewed from line 472 to line 531.
Comment 8: Even though the framework is suggested, it's not clear how it can be used in real life. For example, how can hospitals or training programs actually use these ideas?
Response 8:
Comments 9: The conclusions should detail the specific, measurable outcomes this framework is designed to achieve, such as nurse retention or improved patient outcomes. Provide more active recommendations for researchers and policymakers
Response 9: Conclusion have been revised.
Comments 10: See again referencing style for consistency; at times, "et al." is not included.
Response 10: We have, accordingly revised referencing.
Comments 11: You can also improve language. For example, avoid passive constructions, define terms like "psychological stress", "sufficiently empowered" and make sure that any table, figure, etc. in your paper is well-captioned and described in the text.
Response 11: We have revised the language construction.
|
||
Reviewer 2 Report
Comments and Suggestions for Authors
The article does not meet the criteria for this post type - Article - These are original research manuscripts. The work should inform scientifically based experiments and provide a substantial amount of new information. The article should include the most recent and relevant references in the field. The structure should include the sections Abstract, Keywords, Introduction, Materials and methods, Results, Discussion and Conclusions.
Submitted article of unfulfilled requirements for another type of article review change according to IJERPH criteria - description of methodology (for example, meta-analysis) is missing. Authors can also choose another qualitative method, which must be clearly described and followed when analyzing and interpreting the results.
Comments on the Quality of English Language I have no comments.
Author Response
|
Response to Reviewer 2 Comments
|
||
|
1. Summary |
|
|
|
Thank you very much for taking the time to review this manuscript. Please find the detailed responses below and the corresponding revisions/corrections highlighted/in track changes in the re-submitted files.
|
||
|
2. Questions for General Evaluation |
Reviewer’s Evaluation |
Response and Revisions |
|
Does the introduction provide sufficient background and include all relevant references?
|
Must be improved |
Has been revised |
|
Is the research design appropriate?
|
Must be improved |
Has been revised |
|
Are the methods adequately described?
|
Not Applicable |
|
|
Are the results clearly presented? |
Not Applicable
|
|
|
Are the conclusions supported by the results?
|
Not Applicable |
|
|
|
||
Reviewer #2
The article does not meet the criteria for this post type - Article - These are original research manuscripts. The work should inform scientifically based experiments and provide a substantial amount of new information. The article should include the most recent and relevant references in the field. The structure should include the sections Abstract, Keywords, Introduction, Materials and methods, Results, Discussion and Conclusions.
Submitted article of unfulfilled requirements for another type of article review change according to IJERPH criteria - description of methodology (for example, meta-analysis) is missing. Authors can also choose another qualitative method, which must be clearly described and followed when analyzing and interpreting the results.
Autors' Resonse
Thank you for reviewing the manuscript. The article has been revised as per per attached.The article is the original research study. The reference list is attached on the manuscript. The methodology has been adequately revised.
Reviewer 3 Report
Comments and Suggestions for Authors
Although qualitative studies have not been sufficiently represented in systematic review studies for many years, it has been observed that different methods and techniques have emerged in recent years for the synthesis of qualitative studies. Systematic reviews conducted without complying with the standards cannot be expected to meet the expectations of the readers. The study is important in terms of being a study that completes the deficiency in this regard.
It would be appropriate to support the missing qualitative studies regarding the stages given below. Some studies are given in the suggested framework and some are not.
Stages of Qualitative Evidence Synthesis Systematic Review
1 Development of a clearly formulated systematic review question
2 Determination of the scope of the literature
3 Formal definition of the relevant literature
4 Initial evaluation of study reports
5 Analysis and synthesis
6 Preliminary synthesis
7 Full synthesis
8 Dissemination (Dissemination bias)
9 Throughout the process (Bringing together different perspectives for more diversity and richness of interpretation)
Author Response
|
Response to Reviewer 4 Comments
|
||
|
1. Summary |
|
|
|
Thank you very much for taking the time to review this manuscript. Please find the detailed responses below and the corresponding revisions/corrections highlighted/in track changes in the re-submitted files.
|
||
|
2. Questions for General Evaluation |
Reviewer’s Evaluation |
Response and Revisions |
|
Does the introduction provide sufficient background and include all relevant references?
|
Can be improved |
Has been revised |
|
Is the research design appropriate?
|
Can be improved |
Has been revised |
|
Are the methods adequately described?
|
Can be improved |
Has been revised |
|
Are the results clearly presented? |
Can be improved
|
Has been revised |
|
Are the conclusions supported by the results?
|
Can be improved |
Has been revised |
Reviewer #3
Although qualitative studies have not been sufficiently represented in systematic review studies for many years, it has been observed that different methods and techniques have emerged in recent years for the synthesis of qualitative studies. Systematic reviews conducted without complying with the standards cannot be expected to meet the expectations of the readers. The study is important in terms of being a study that completes the deficiency in this regard.
It would be appropriate to support the missing qualitative studies regarding the stages given below. Some studies are given in the suggested framework and some are not.
Stages of Qualitative Evidence Synthesis Systematic Review
1 Development of a clearly formulated systematic review question
2 Determination of the scope of the literature
3 Formal definition of the relevant literature
4 Initial evaluation of study reports
5 Analysis and synthesis
6 Preliminary synthesis
7 Full synthesis
8 Dissemination (Dissemination bias)
9 Throughout the process (Bringing together different perspectives for more diversity and richness of interpretation)
Authors ' response
Thank you for reviewing the manuscript. The manuscript is a qualitative study and has been revised as per attached. The manuscript is not a systematic review article and did not follow the review format.